CERN-TH-2025-012, IPPP/25/09, ZU-TH 11/25

# NNLOJET: a parton-level event generator for jet cross sections at NNLO QCD accuracy

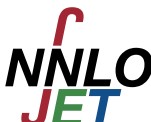

**NNLOJET Collaboration**

A. Huss[1,⋆], L. Bonino[2], O. Braun-White[3], S. Caletti[4], X. Chen[5], J. Cruz–Martinez[1],
J. Currie[3], W. Feng[2], G. Fontana[2], E. Fox[3], R. Gauld[6], A. Gehrmann–De Ridder[2,4],
T. Gehrmann[2], E.W.N. Glover[3], M. Höfer[7], P. Jakubčík[2], M. Jaquier[8], M. Löchner[2],
F. Lorkowski[2], I. Majer[4], M. Marcoli[3], P. Meinzinger[2], J. Mo[2], T. Morgan[3], J. Niehues[3,9],
J. Pires[10,11], C. T. Preuss[12,13], A. Rodriguez Garcia[4], K. Schönwald[2], R. Schürmann[2],
V. Sotnikov[2], G. Stagnitto[14], D. Walker[3], S. Wells[3], J. Whitehead[15], T.Z. Yang[2] and
H. Zhang[16]

**1** Theoretical Physics Department, CERN, 1211 Geneva 23, Switzerland
**2** Physik-Institut, Universität Zürich, Winterthurerstrasse 190, 8057 Zürich, Switzerland
**3** Institute for Particle Physics Phenomenology, Department of Physics,
University of Durham, Durham, DH1 3LE, UK
**4** Institute for Theoretical Physics, ETH, CH-8093 Zürich, Switzerland
**5** School of Physics, Shandong University, Jinan, Shandong 250100, China
**6** Max-Planck-Institut für Physik, Werner-Heisenberg-Institut, 85748 Garching, Germany
**7** Institute for Theoretical Physics, KIT, 76131 Karlsruhe, Germany
**8** Institute for Theoretical Particle Physics, KIT, 76131 Karlsruhe, Germany
**9** EKFZ for Digital Health, TU Dresden, 01307 Dresden, Germany
**10** LIP, Avenida Professor Gama Pinto 2, 1649-003 Lisboa, Portugal
**11** Faculdade de Ciências, Universidade de Lisboa, 1749-016 Lisboa, Portugal
**12** Department of Physics, University of Wuppertal, 42119 Wuppertal, Germany
**13** Institut für Theoretische Physik, Georg-August-Universität Göttingen,
37077 Göttingen, Germany
**14** Università degli Studi di Milano-Bicocca and INFN,
Piazza della Scienza 3, 20126 Milano, Italy
**15** Jagiellonian University, ul. prof. Stanislawa Łojasiewicza 11, 31-342 Kraków, Poland
**16** PSI Center for Neutron and Muon Sciences, 5232 Villigen PSI, Switzerland

⋆ alexander.huss@cern.ch

## Abstract

The antenna subtraction method for NNLO QCD calculations is implemented in the NNLO-JET parton-level event generator code to compute jet cross sections and related observables in electron–positron, lepton–hadron and hadron–hadron collisions. We describe the open-source NNLOJET code and its usage.

# 1 Introduction

The Standard Model of particle physics is firmly established as the theoretical framework describing the interaction of elementary particles up to the highest energy scales that are accessible at particle collider experiments. Increasingly accurate experimental measurements allow to test the Standard Model and to determine its parameters at an unprecedented level of quantitative precision. To enable these precision physics studies, a close interplay between experimental measurements and theoretical predictions is required.

Many precision observables are derived from hadronic final states, typically in the form of identified jets, which can be related to the production of quarks and gluons (partons) in hard scattering processes [1, 2]. These jet observables provide clean probes of the dynamics of the strong interaction (quantum chromodynamics, QCD) at electron–positron colliders and they affect all types of precision measurements at lepton–hadron or hadron–hadron colliders due to the partonic structure of the colliding hadrons. The theoretical description of these observables must take account of the detailed definition of the final state that is applied in the experimental measurement by incorporating the jet clustering algorithm and any fiducial cuts and isolation requirements on the final-state objects. The resulting predictions for fiducial cross sections can then be compared directly to the experimental data, thereby they stress-test the Standard Model and refine the determination of the parameters (Standard Model couplings and masses, parton distribution functions) that enter the theoretical predictions.

Theoretical predictions rely on a perturbation theory expansion in the coupling constants. In the case of hadron-collider processes and jet observables, QCD corrections are particularly important due to the large value of the strong coupling constant $\alpha_s$ compared to the other Standard Model interactions. At higher orders in perturbation theory, a Born-level process with given final-state multiplicity receives two types of contributions: virtual corrections from closed particle loops in the underlying scattering amplitudes and real-radiation corrections from final states with higher multiplicity. At next-to-leading order (NLO), these correspond to one-loop virtual corrections (V) and single real emission corrections (R). With increasing orders, corrections can be of mixed type, with next-to-next-to-leading order (NNLO) consisting of two-loop double virtual (VV), mixed one-loop single real emission (RV) and tree-level double-real emission corrections (RR) and so on. Each individual subprocess contribution contains infrared singularities arising from soft or collinear momentum configurations in loop integrals and real-radiation processes; only the sum of all subprocesses contributing to a well-defined (infrared-safe, [3]) final state is finite and physically meaningful. These infrared singularities are explicit (as poles in the dimensional regularization parameter) only in the virtual loop amplitudes [4–7], while leading to divergent phase-space integrals in the real-radiation processes. To obtain numerically finite predictions from real-radiation corrections, their infrared singularity structure must be made explicit.

For this task, various infrared subtraction methods have been proposed and implemented for NLO calculations [8–10]. When combined with generic procedures for the evaluation of one-loop virtual amplitudes [11–14] for any process, these methods have enabled the full automation of NLO corrections in QCD and the electroweak theory, becoming part of multi-purpose event simulation programs [15–19], which are complemented by the dedicated libraries for NLO processes in MCFM [20] and VBFNLO [21]. These perturbative fixed-order calculations are implemented as parton-level Monte Carlo event generation programs. Such a program contains the infrared-finite remainders of all parton-level subprocesses. These are evaluated over their respective phase spaces of different multiplicities, producing weighted parton-level events with full kinematic information that can be analysed according to the experimental fiducial cross section definition.

Building upon the universal factorization of QCD real-radiation amplitudes in their in-

frared limits [22–26], several NNLO subtraction methods have been formulated [27–38]. These methods have enabled the calculation of NNLO QCD corrections to a large number of collider processes [39] with up to three identified final-state objects. Such calculations are performed on a case-by-case basis in the framework of parton-level Monte Carlo event generation, typically requiring high-performance computing resources to yield stable and numerically reliable results. The resulting codes often require dedicated tuning and post-processing for each observable. They are usually not broadly available as open-source. Notable exceptions are EERAD3 [40, 41] for jet production in electron-positron annihilation, the MATRIX [42] and MiNNLOPS [43, 44] codes for processes with colour-neutral or heavy-quark final states, the NNLOCAL code [45] for colour-neutral final states and the NNLO-upgrade of the MCFM code [20, 46] for selected di-boson processes.

In this work, we present the open-source release of the NNLOJET parton-level event generator code, which employs the antenna subtraction method [30, 47, 48] to compute the NNLO QCD corrections to several classes of jet production processes in hadron–hadron, lepton–hadron and electron–positron collisions. In the following, we review the antenna subtraction method in Sect. 2, describe the NNLOJET installation and general usage in Sect. 3 and the currently available processes in Sect. 4. The detailed usage of NNLOJET is documented through its runcard syntax and process workflow in Sections 5–6. The NNLOJET code is available on HEPForge: http://nnlojet.hepforge.org and is distributed under the GNU General Public License v3.0.

## 2   Antenna subtraction

The differential cross section $d\sigma$ for a specific final state at a hadron collider is obtained by the convolution of parton distributions $f_{i|h}$ with parton-level cross sections $d\hat{\sigma}$

$$d\sigma = \sum_{a,b} d\hat{\sigma}_{ab} \otimes f_{a|h_1} \otimes f_{b|h_2} \,. \tag{1}$$

These parton-level cross sections are computed in perturbative QCD as an expansion in the strong coupling:

$$d\hat{\sigma}_{ab} = d\hat{\sigma}_{ab,\text{LO}} + d\hat{\sigma}_{ab,\text{NLO}} + d\hat{\sigma}_{ab,\text{NNLO}} + \dots \,. \tag{2}$$

Their computation must account for the definition of the final state (jet and object reconstruction as well as fiducial cuts on the final-state objects) and allow the extraction of differential distributions in all desired kinematic variables. This flexibility can only be attained by analysing all parton-level contributions to the cross section at a given perturbative order at the level of the respective partonic kinematics. The resulting implementation is a parton-level event generator, which can compute the differential cross sections for any infrared-safe observable that can be derived from a given Born-level process.

The leading-order contribution $d\hat{\sigma}_{ab,\text{LO}}$ defines this Born-level process for $n$ final-state objects:

$$d\hat{\sigma}_{ab,\text{LO}} = \int_n d\hat{\sigma}_{ab}^B \,, \tag{3}$$

where $d\hat{\sigma}_{ab}^B$ is the Born-level differential cross section including the $n$-object final-state definition and $\int_n$ indicates the integration over the $n$-particle final-state phase space. At LO, each final-state object is described by a single particle, such that the event reconstruction amounts

to a simple check that all final-state objects are non-overlapping in phase space and are all within the fiducial acceptance.

The NLO contribution

$$
d\hat{\sigma}_{ab,\text{NLO}} = \int_n \left( d\hat{\sigma}^V_{ab,\text{NLO}} + d\hat{\sigma}^{MF}_{ab,\text{NLO}} \right) + \int_{n+1} d\hat{\sigma}^R_{ab,\text{NLO}}
\tag{4}
$$

consists of virtual one-loop corrections $d\hat{\sigma}^V$, real-radiation corrections $d\hat{\sigma}^R$, and mass-factorization counterterms $d\hat{\sigma}^{MF}$ associated with unresolved radiation from the incoming partons. All three contributions are separately infrared-divergent and only their sum is infrared-finite and physically well-defined. The singularities are commonly made explicit by working in dimensional regularization in $d = 4 - 2\epsilon$ dimensions.

At NNLO, the cross section consists of double-virtual (two loop and one-loop squared) corrections $d\hat{\sigma}^{VV}$, mixed real–virtual corrections $d\hat{\sigma}^{RV}$, and double-real-radiation corrections $d\hat{\sigma}^{RR}$, as well as mass-factorization terms $d\hat{\sigma}^{MF,1}$ and $d\hat{\sigma}^{MF,2}$ associated with single and double initial-state radiation:

$$
\begin{aligned}
d\hat{\sigma}_{ab,\text{NNLO}} = {} & \int_n \left( d\hat{\sigma}^{VV}_{ab,\text{NNLO}} + d\hat{\sigma}^{MF,2}_{ab,\text{NNLO}} \right) \\
& + \int_{n+1} \left( d\hat{\sigma}^{RV}_{ab,\text{NNLO}} + d\hat{\sigma}^{MF,1}_{ab,\text{NNLO}} \right) + \int_{n+2} d\hat{\sigma}^{RR}_{ab,\text{NNLO}} \,.
\end{aligned}
\tag{5}
$$

All five contributions are infrared-divergent; the cancellation of infrared singularities is accomplished only if they are all summed together.

The infrared singularities at NLO and NNLO are of two forms: explicit $\epsilon$-poles from loop integrals in the virtual corrections [4] and implicit soft and/or collinear singularities in the real corrections [22–26], which become explicit only upon integration over the respective phase space. Consequently, it is not possible to implement equations (4) and (5) directly in a numerical code. To accomplish the cancellation of infrared singularities among the different contributions at NLO and NNLO, one introduces a subtraction procedure that extracts the implicit infrared singularities from real-radiation processes, makes them explicit by phase-space integration and combines them with the virtual corrections. The resulting expressions read

$$
\begin{aligned}
d\hat{\sigma}_{ab,\text{NLO}} = {} & \int_{n+1} \left( d\hat{\sigma}^R_{ab,\text{NLO}} - d\hat{\sigma}^S_{ab,\text{NLO}} \right) \\
& + \int_n \left( d\hat{\sigma}^V_{ab,\text{NLO}} - d\hat{\sigma}^T_{ab,\text{NLO}} \right)
\end{aligned}
\tag{6}
$$

and

$$
\begin{aligned}
d\hat{\sigma}_{ab,\text{NNLO}} = {} & \int_{n+2} \left( d\hat{\sigma}^{RR}_{ab,\text{NNLO}} - d\hat{\sigma}^S_{ab,\text{NNLO}} \right) \\
& + \int_{n+1} \left( d\hat{\sigma}^{RV}_{ab,\text{NNLO}} - d\hat{\sigma}^T_{ab,\text{NNLO}} \right) \\
& + \int_n \left( d\hat{\sigma}^{VV}_{ab,\text{NNLO}} - d\hat{\sigma}^U_{ab,\text{NNLO}} \right) \,.
\end{aligned}
\tag{7}
$$

The subtraction terms relate among each other as follows:

$$
\begin{aligned}
\mathrm{d}\hat{\sigma}_{ab,\mathrm{NLO}}^{T} &= -\int_{1} \mathrm{d}\hat{\sigma}_{ab,\mathrm{NLO}}^{S} - \mathrm{d}\hat{\sigma}_{ab,\mathrm{NLO}}^{MF}\,, \\
\mathrm{d}\hat{\sigma}_{ab,\mathrm{NNLO}}^{S} &= \mathrm{d}\hat{\sigma}_{ab,\mathrm{NNLO}}^{S,1} + \mathrm{d}\hat{\sigma}_{ab,\mathrm{NNLO}}^{S,2}\,, \\
\mathrm{d}\hat{\sigma}_{ab,\mathrm{NNLO}}^{T} &= \mathrm{d}\hat{\sigma}_{ab,\mathrm{NNLO}}^{T,1} - \int_{1} \mathrm{d}\hat{\sigma}_{ab,\mathrm{NNLO}}^{S,1} - \mathrm{d}\hat{\sigma}_{ab,\mathrm{NNLO}}^{MF,1}\,, \\
\mathrm{d}\hat{\sigma}_{ab,\mathrm{NNLO}}^{U} &= -\int_{1} \mathrm{d}\hat{\sigma}_{ab,\mathrm{NNLO}}^{T,1} - \int_{2} \mathrm{d}\hat{\sigma}_{ab,\mathrm{NNLO}}^{S,2} - \mathrm{d}\hat{\sigma}_{ab,\mathrm{NNLO}}^{MF,2}\,.
\end{aligned}
\tag{8}
$$

Here $\int_{i}$ denotes the integration over the phase space associated with $i$ unresolved partons and the decomposition into $\mathrm{d}\hat{\sigma}^{S,(1,2)}$ separates single- and double-unresolved configurations. By introducing these subtraction terms, each line in equations (6) and (7) is free of implicit and explicit singularities, thus allowing a numerical implementation in a parton-level event generator.

NNLOJET implements the antenna subtraction method [30,47,48], which derives the subtraction terms starting from colour-ordered decompositions of all real-radiation contributions. The colour-ordering allows identifying each unresolved real-radiation configuration with a pair of hard radiator partons, which emit the unresolved partons. This configuration is then described by an antenna function consisting of the hard radiators and unresolved radiation in between them. Antenna functions were derived at NLO and NNLO for quark–antiquark [49], quark–gluon [50] and gluon–gluon [51] pairs as hard radiators. All quarks are treated as massless. The unintegrated subtraction terms are obtained from products of antenna functions and reduced matrix elements of lower partonic multiplicity, which are summed over all configurations of hard radiators and unresolved partons. The kinematics of the antenna functions and the reduced matrix elements are factorized from each other through a phase-space mapping [47,52], such that the antenna function can be integrated over the antenna phase space of the hard radiator partons and the unresolved radiation while leaving the kinematics of the reduced matrix element unaffected. The integrated antenna functions enter the subtraction terms $\mathrm{d}\hat{\sigma}^{T}$ and $\mathrm{d}\hat{\sigma}^{U}$ where they establish the explicit cancellation of infrared $\epsilon$-poles between real and virtual contributions. They were computed up to NNLO for all different configurations of the hard radiators: final–final [30], initial–final [53] and initial–initial [54–56].

With these ingredients, all terms in equation (8) can be implemented in the antenna subtraction formalism, as described in detail in [48]. For all processes available in NNLOJET, this implementation has been made on a process-by-process basis, usually starting from the implicit infrared singularity structure of the double-real-radiation process $\mathrm{d}\hat{\sigma}^{RR}$. A fully automated construction of the antenna subtraction terms in colour space is currently under development [57,58].

## 3 The NNLOJET framework

The NNLOJET code is a parton-level event generator that provides the framework for the implementation of jet production processes to NNLO accuracy in QCD, using the antenna subtraction method. It contains the event generator infrastructure (Monte Carlo phase-space integration, event handling and analysis routines) and provides the unintegrated and integrated antenna functions and the phase-space mappings for all kinematic configurations. The NNLOJET phase-space integration is based on optimized parametrizations of the respective subprocess phase spaces (described in detail in [59]), which are then integrated using the adaptive Monte Carlo routine VEGAS [60].

The implementation of processes in the NNLOJET framework requires the availability of the matrix elements for all RR, RV, and VV contributions, as well as the construction of the antenna subtraction terms. The subtraction terms have been validated by verifying their pointwise convergence to the respective subprocess matrix elements in all limits, as described in detail in [57, 61]. Some of the one-loop subprocess matrix elements were taken from the MCFM code [20], in particular for processes involving vector bosons [62]. The two-loop virtual corrections to all processes with four-point kinematics or beyond contain various special functions, in particular harmonic polylogarithms [63] and their generalizations. These are evaluated using HPLOG [64], TDHPL [65] and CHAPLIN [66], which are all included as part of the NNLOJET distribution. The evaluation of parton distributions and of the strong coupling constant is performed using the standard LHAPDF [67] interface.

## 3.1 External dependencies

The NNLOJET code is mainly written in modern Fortran with some modules, dependencies and driver files written in C++ and Python. Multi-threading capabilities are supported through OPENMP. The NNLOJET installation requires compilers for Fortran/C/C++, a Python 3 interpreter (minimum version 3.10), as well as CMake (minimum version 3.18) to be available on the system. In addition, LHAPDF 6 [67] must be present, ideally with the `lhapdf-config` executable searchable from the `$PATH` of the system. The user is responsible for downloading the necessary PDF sets to be used in the calculation.

## 3.2 Installation

The source files can be downloaded from https://nnlojet.hepforge.org as a tarball, which unpacks into a directory `nnlojet-X.Y.Z`. The following commands are then used to build NNLOJET:

```
$ cd nnlojet-X.Y.Z
$ mkdir build
$ cd build
$ cmake .. [options]
$ make [-jN]
$ make install
```

The compilation can be run on `N` cores in parallel with the `-jN` optional argument. The installation can be customised, by passing additional `[options]` to the `cmake` command. The syntax is `-D <var>=<value>`. The most relevant options are:

- `CMAKE_INSTALL_PREFIX=/path/to/nnlojet`: target directory where to install NNLOJET (default: `/usr/local`);

- `CMAKE_BUILD_TYPE=Debug|Release|RelWithDebInfo|MinSizeRel`: build option for NNLOJET (default: `Release`);

- `LHAPDF_ROOT_DIR=/path/to/LHAPDF`: CMake will attempt to find the library automatically but the path can be supplied manually;

- `Python3_ROOT_DIR=/path/to/python`: CMake will attempt to find a python installation automatically but the path can be supplied manually;

- `DOKAN=ON|OFF`: optionally skip the installation of the `nnlojet-run` script (default: `ON`);

- `OPENMP=ON|OFF`: enable OpenMP support (default: `OFF`).

After successful completion of the build and installation, the NNLOJET executable will be located at `/path/to/nnlojet/bin/NNLOJET`. It is recommended to add the location of the NNLOJET `bin` directory to the system `$PATH` variable.

For system-dependent settings (e.g. compiler preferences and options) the user should consult the cmake documentation.

## 3.3   Usage of NNLOJET

The recommended running mode of NNLOJET is through the Python workflow described in Section 6, which will launch the NNLOJET executable. The process selection, parameter input and output control of NNLOJET are steered through a plain-ASCII runcard file. With this file, NNLOJET can also be launched manually from the command line:

```
$ NNLOJET --run <runcard>
```

The NNLOJET executable further offers the optional command-line flags:

```
--help             -  display command line options
--iseed            -  override the seed number of the runcard
--imember          -  override the PDF member of the runcard
--listprocs        -  query all available processes
--listobs <process>  -  display list of valid observables
```

NNLOJET uses dynamical libraries to load individual processes and initializes many parts of the computational setup at runtime. As a result, the process to be considered, observable selectors and histograms can be specified directly in the runcard in a flexible way without requiring the re-compilation of the code. Any dimensionful parameter is given in units of [GeV]. Cross sections outputs are in [fb].

Complete examples for runcards are provided together with reference output on the NNLO-JET webpage [https://nnlojet.hepforge.org/examples.html](https://nnlojet.hepforge.org/examples.html).

# 4   Processes in NNLOJET

The NNLOJET distribution under [https://nnlojet.hepforge.org](https://nnlojet.hepforge.org) will be updated on a regular basis, especially by adding new processes. The list of available processes in the current version can be obtained by

```
$ NNLOJET --listprocs
```

For version `v1.0.0` this yields:

```
$ NNLOJET --listprocs
Available processes up to NNLO accuracy for hadron-hadron colliders:
*    Z      : production of a Z-boson;
*    ZJ     : production of a Z-boson plus a jet;
*    WP     : production of a positively charged W-boson;
*    WPJ    : production of a positively charged W-boson plus a jet;
*    WM     : production of a negatively charged W-boson;
*    WMJ    : production of a negatively charged W-boson plus a jet;
*    H      : production of a H-boson;
*    HJ     : production of a H-boson plus a jet;
*    H2     : production of a H-boson with decay to two colour singlets, in particular:
*     - HTO2P    : decay to two photons;
*    H2J    : production of a H-boson plus a jet with decay to two colour singlets, in particular:
*     - HTO2PJ   : decay to two photons;
*    H3     : production of a H-boson with decay to three colour singlets, in particular:
*     - HTO2L1P  : Dalitz decay to (l+l-) and photon;
*    H3J    : production of a H-boson plus a jet with decay to three colour singlets, in particular:
*     - HTO2L1PJ : Dalitz decay to (l+l-) and photon;
*    H4     : production of a H-boson with decay to four colour singlets, in particular:
*     - HTO4E    : decay to (l+l-) (l+l-);
*     - HTO2E2MU : decay to (l1+l1) (l2+l2-);
*     - HTO2L2N  : decay to (l1-nu1b) (l2+nu2);
*    H4J     : production of a H-boson plus a jet with decay to four colour singlets, in particular:
*     - HTO4EJ   : decay to (l+l-) (l+l-);
*     - HTO2E2MUJ : decay to (l1+l1) (l2+l2-);
*     - HTO2L2NJ  : decay to (l1-nu1b) (l2+nu2);
```

```
*    GJ      : production of a photon plus a jet;
*    GG      : production of two photons;
*    JJ      : production of two jets;
Available processes up to NNLO accuracy for electron-positron colliders:
*    eeJJ    : production of two jets;
*    eeJJJ   : production of three jets;
Available processes up to NNLO accuracy for lepton-hadron colliders:
*    epLJ    : production of a jet through neutral-current exchange;
*    epLJJ   : production of two jets through neutral-current exchange;
Available processes up to NNLO accuracy for electron-hadron colliders:
*    epNJ    : production of a jet through charged-current exchange;
*    epNJJ   : production of two jets through charged-current exchange;
Available processes up to NNLO accuracy for positron-hadron colliders:
*    epNbJ   : production of a jet through charged-current exchange;
*    epNbJJ  : production of two jets through charged-current exchange;
```

For each process type available at NNLO QCD, it is implicitly understood that NNLOJET also provides the NLO correction to the process with one additional jet and the LO of the process with two additional jets. For example, since $Z+1j$ is available at NNLO, $Z+2j$ can be computed at NLO and $Z+3j$ at LO. To perform NLO and LO calculations of processes with additional jets in the final state, one can simply require a higher number of resolved jets in the runcard file. For example, the NLO correction to $Z + 2j$ can be computed via an NNLO calculation for the ZJ process requiring at least two jets.

The processes included in the current release are described in the following. Details on their NNLOJET implementation and on validation tests are described in detail in the respective publications. In general, the following checks have been applied on all process: (1) point-tests [57, 61] and (2) technical-cut checks for RR and RV contributions; (3) finiteness of the RV and VV contributions.

In the point tests, multiple phase space points are generated near each potentially singular infrared limit, with the proximity to the limit being controlled by a one-dimensional parameter. It is then validated that the ratio of the matrix element and subtraction term approaches unity in each limit and that the distribution of the ratios is progressively concentrated in a tighter interval around unity as the control parameter decreases.

The phase space generation in all processes imposes a lower technical cut on all Mandel-stam invariants, which is chosen considerably smaller than the smallest scale that is resolved by the final-state definition. For a properly functioning subtraction at RR and RV level, the deep-infrared region in the vicinity of the technical cut gives a negligible contribution to the cross section, which is checked by its independence on the technical cut.

The pole tests are performed numerically for a large number of bulk phase space points on the RV process, where $\epsilon$-poles of matrix elements, mass-factorization terms and subtraction terms are implemented explicitly. In the VV process, all integrated subtraction terms are combined with the mass-factorization terms and the pole terms extracted [4] from the double-virtual matrix element to obtain analytical pole cancellation and to define the finite remainder that is implemented numerically.

## 4.1 Jet production in electron–positron collisions

Jet final states in e$^+$e$^-$ annihilation and related event shapes played a key role in establishing QCD as theory of the strong interaction and their theoretical study has been a major driver of developments in precision calculations in QCD. The e$^+$e$^- \to 3j$ cross section and related event shapes were the first jet observables to be computed to NNLO in QCD [68], originally implemented in the code EERAD3 [40] which is also based on antenna subtraction. The implementation of e$^+$e$^- \to 2j$ and e$^+$e$^- \to 3j$ in NNLOJET is described in [69]. It uses the matrix elements for vector-boson decay into partons [70–76], which account for the incoming lepton kinematics. The commonly used LEP-era event-shape variables and jet algorithms [77] are part of the generic NNLOJET infrastructure.

## 4.2 Jet production in electron–proton collisions

Jet production in deeply inelastic electron–proton scattering (DIS) allows to probe aspects of the proton structure that are difficult to assess in inclusive DIS and provides important precision tests of QCD. The implementation of ep → lepton + 2$j$ up to NNLO QCD is described in [78, 79], including both neutral-current and charged-current DIS. It is based on crossings [75] of the e$^+$e$^-$ → jets matrix elements to DIS kinematics. Jets can be defined in the Breit frame or in the electron–proton collider frame and DIS event shapes [80] are part of the NNLOJET infrastructure.

## 4.3 Jet production in hadronic collisions

Jet production is the most ubiqutous QCD process at hadron colliders, probing the basic 2 → 2 parton scattering reactions. A number of kinematic distributions can be derived from single-inclusive jet production pp → $j$ + X and from dijet production pp → 2$j$. Their NNLOJET implementation is described in detail in [61, 81–87]. It is based on the relevant subprocess matrix elements up to NNLO [88–100]. The generic NNLOJET infrastructure allows selecting various different jet algorithms, performing scans over jet-resolution parameters and the setting of renormalization and factorization scales [101] either in a jet-based or event-based manner.

## 4.4 Vector-boson-plus-jet production

Precision studies of the electroweak interaction are commonly performed through massive vector-boson production, identified through their leptonic decay mode. Further kinematic information can be obtained by considering vector-boson-plus-jet final states. Observables derived from vector-boson and vector-boson-plus-jet production are widely used in precision measurements of electroweak parameters and of the proton structure. The NNLOJET implementation for pp → ($\gamma^*$, Z) + 0$j$, pp → ($\gamma^*$, Z) + 1$j$ is described in [102–108] and for pp → W + 0$j$, pp → W + 1$j$ in [106, 109]. The implementations are based on parton-level matrix elements up to NNLO, obtained as kinematic crossings [76] of the e$^+$e$^-$ → jets matrix elements. Decays into lepton pairs and the off-shell vector-boson propagator are included. It is also possible to obtain simplified predictions for on-shell vector-boson production in the narrow-width prescription.

## 4.5 Photon-plus-jet and di-photon production

The production of isolated photons is a classical collider observable. Its theoretical description must account for the experimental prescription that is used to isolate the photons from hadronic activity in the event. This prescription is typically formulated in terms of an isolation cone around the photon direction, which admits only a limited amount of hadronic energy. All relevant experimental measurements use a fixed-size cone, while theory calculations also consider dynamical isolation cones [110], which gradually reduce the allowed hadronic energy to zero towards the photon direction in the cone centre. Only fixed-size cones admit some amount of collinear photon radiation off QCD partons, thus inducing a dependence on photon fragmentation processes [111]. The NNLOJET implementation of pp → $\gamma$ + X and pp → $\gamma$ + 1$j$ to NNLO QCD with a dynamical isolation cone is described in [112]. Its extension to fixed-cone isolation is documented in [113], employing an empirical parametrization of the photon fragmentation functions [114]. Di-photon production pp → $\gamma\gamma$ is currently implemented to NNLO QCD only for dynamical cone isolation [115]. The matrix elements up to NNLO QCD for both processes are taken from [116–119].

### 4.6 Higgs-boson-plus-jet production

The dominant Higgs-boson-production process at the LHC is gluon fusion, which is mediated through a virtual top-quark loop. The NNLOJET implementation of $pp \to H + 0j$ and $pp \to H + 1j$ to NNLO QCD is described in [120]. It is based on an effective field theory (EFT) which couples the Higgs-boson to the gluon field strength, resulting from integrating out the top-quark loop in the infinite-mass limit [121, 122]. The relevant matrix elements up to NNLO are based on [123–127]. The Higgs boson is considered to be on-shell and a narrow-width approximation is used. Besides the on-shell Higgs production, the Higgs decay modes to $2\gamma$ [128], $4\ell$ and $2\ell 2\nu$ [129] as well as to $2\ell + \gamma$ [130] are implemented, allowing arbitrary fiducial cuts on the Higgs decay products. Heavy-quark mass corrections are implemented at LO QCD for $pp \to H + 0j$ and $pp \to H + 1j$, allowing for a multiplicative reweighting of the infinite-mass EFT predictions.

## 5 Runcard syntax

### 5.1 General syntax

The general syntax of a runcard file follows these rules:

- Each statement should be written on a *separate* line. Wrapping across multiple lines requires the line continuation symbol `&` at the end of the line and at the beginning of the following line.

- Comments start with an exclamation mark `!`, all characters that follow `!` will be ignored by the parser.

- The parser does not distinguish between upper- and lower-case letters, with the exception of observable names (see below) which are case sensitive but typically defined in lower case.

- Assignments follow the syntax `<var>` = `<val>`[`...`] with any white space allowed in between and possibly extra options provided inside square brackets [`...`].

- Lists are specified as [`<val1>`,`<val2>`,`...`,`<valN>`] and white space will be ignored.

- The types of literals follow Fortran syntax:

    - `BOOL`: boolean variables, `.true.` or `.false.`;
    - `INT`: integers;
    - `DBLE`: double precision variables, e.g. `13.6d3` and `1d-8`;
    - `STR`: string expression made up with any ASCII characters (without any quotes).

The runcard is composed of the following separate blocks:

- `PROCESS` [**required**]: defines and specifies the process;

- `RUN` [**required**]: defines the technical parameters and settings;

- `PARAMETERS`: defines the input settings and parameter values;

- `SELECTORS` [**required**]: defines the event and object selectors for the fiducial cross section;

- `HISTOGRAMS`: defines output histograms (observable, binning, extra options);

- `SCALES` [**required**]: specifies the set of renormalization and factorization scales for simultaneous evaluation;

- `CHANNELS` [**required**]: selects the subprocesses to be activated;

as well as optional single-line settings:

- `REWEIGHT`: weight to bias the sampling of the integrator;

- `PDF_INTERPOLATION`: switch on dynamical interpolation for PDFs.

They are described in detail in Sect. 5.3.

## 5.2 Observables

The event selectors for the fiducial cross section, settings for the histograms and values for the scales are specified in terms of observables. The list of available observables depends on the selected process.

Each observable is implemented in a generic way and is internally assigned a unique identifier. Dimensionful observables are defined in [GeV] units. In order to obtain a list of all available observables for a given process with a short description one simply needs to run `NNLOJET --listobs <process_name>` in the terminal. For example, for Z-plus-jet production, it returns:

```
$ NNLOJET --listobs ZJ
*
*    > jet acceptance cuts <
*
*    jets_pt
*    jets_y
*    jets_abs_y
*    jets_eta
*    jets_abs_eta
*    jets_et
*    jets_min_dr_lj
*
*     > process-specific observables <
*
*    ptlm        --- transverse momentum of l-
*    Elm         --- energy of l-
*    ylm         --- (pseudo-)rapidity of l-
*    abs_ylm     --- |ylm|
*    ptlp        --- transverse momentum of l+
*    Elp         --- energy of l+
*    ylp         --- (pseudo-)rapidity of l+
*    abs_ylp     --- |ylp|
*    ptl1        --- transverse momentum of leading lepton
*    yl1         --- (pseudo-)rapidity of leading lepton
*    abs_yl1     --- |yl1|
...
```

Internally, each observable has an attribute to query its validity for an event it was evaluated for. The validity state is typically concerned with observables that are defined based on objects that can vary in multiplicity, such as jets. For example, the transverse momentum of the sub-leading jet in the event `ptj2` is only valid if the event contains at least two jets

after applying all event and jet selection cuts. For a process that only requires one jet to be resolved, e.g. Z + jet, this observable can thus become invalid. While the observable in this case is technically of one order lower in the perturbative accuracy, it is still possible to evaluate it. It is important to keep in mind the validity state of an observable when defining selectors and histograms, as will be discussed below.

### 5.2.1 Reweighting functions

Reweighting functions define multiplicative factors that are applied to the event weight within a limited context. They can be used in different parts of the runcard to influence various behaviours of the run or the output, as described in Sects. 5.3.5 and 5.3.8. The syntax for reweighting functions is:

```
<observable> ** <power> * <factor> + <offset>
```

- The `<observable>` is specified using the unique string identifier;

- The `<power>`, `<factor>` and `<offset>` are `DBLE`;

- The symbols **, * and + can only appear *once* with any white space ignored;

- The ordering of variables and observables does not matter.

For example, all the following lines constitute valid reweighting functions:

```
2            ! a constant factor
ptz          ! an observable
ptz**2       ! an observable raised to a power
3*ptz        ! a rescaled observable
ptz+1        ! an observable with an offset
3*ptz**2+1   ! any combination of the above
```

## 5.3 Runcard sections

In the following, the runcard input is illustrated in detail. Mandatory settings are highlighted as "[**required**]".

### 5.3.1 Process block

The `PROCESS` block defines the process to be calculated and is specified following the general syntax:

```
PROCESS <process_name>
  <option1> = <value1>[<parameters1>]
  <option2> = <value2>[<parameters2>]
  ...
END_PROCESS
```

The currently supported process names (`<process_name>`) at NNLO can be obtained with the `NNLOJET --listprocs` command. As described in Sect. 4, they are:

- `Z`, `ZJ`: production of Z-boson (plus up to one jet), including Z-boson decay to two charged leptons;

- `Wp`, `Wm`, `WpJ`, `WmJ`: production of $W^+/W^-$-boson (plus up to one jet), including W-boson decay to a charged lepton and a neutrino;

- `H`, `HJ`: production of H-boson (plus up to one jet), excluding H-boson decay;

- `H2`, `H2J`: production of H-boson (plus up to one jet) with decay mode to two colour singlets. Specifically:

    - `HTO2P`, `HTO2PJ` with decay mode to a $\gamma$ pair;

- `H3`, `H3J`: production of H-boson (plus up to one jet) with decay mode to three colour singlets. Specifically:

    - `HTO2L1P`, `HTO2L1PJ` with Dalitz decay mode to $\ell^+ \ell^- + \gamma$;

- `H4`, `H4J`: production of H-boson (plus up to one jet) with decay mode to four colour singlets. Specifically:

    - `HTO4E`, `HTO4EJ` with decay mode to two light lepton pairs with the same flavour;
    - `HTO2E2MU`, `HTO2E2MUJ` with decay mode to two light lepton pairs with different flavour;
    - `HTO2L2N`, `HTO2L2NJ` with decay mode to a lepton plus a neutrino pair mediated by $W$-bosons;

- `GJ`: production of photon plus one jet;

- `GG`: production of photon pair;

- `JJ`: production of two jets;

- `eeJJ`, `eeJJJ`: production of up to three jets in electron–position colliders;

- `epLJ`, `epLJJ`: production of up to two jets in neutral-current lepton-proton collisions, resulting in lepton-plus-jets final states (no distinction is made between leptons and antileptons);

- `epNJ`, `epNJJ`: production of up to two jets in charged-current lepton-proton collisions, resulting in neutrino-plus-jets final states;

- `epNbJ`, `epNbJJ`: production of up to two jets in charged-current antilepton-proton collisions, resulting in antineutrino-plus-jets final states.

The process names refer to the underlying Born-level process, which can then be used to compute derived quantities with the same final-state kinematics but without identified jets. For example, `ZJ` also allows the computation of the transverse momentum distribution of the Z-boson, with the jet requirement being replaced by a non-zero transverse momentum of the Z-boson due to partonic recoil. Likewise, `GJ` contains inclusive photon production, `JJ` single-inclusive-jet production, and `eeJJJ` the three-jet-like event shapes.

The currently supported options and values for the `PROCESS` block are listed in the following.

A collider choice is specified with:

- `collider` = `STR` [**required**]: using pre-defined collider setups:

    - proton–proton collider: `PP`;
    - proton–antiproton collider: `PAP`;
    - proton–electron collider (deep-inelastic scattering): `PE`;
    - electron–positron collider: `EPEM`;

- beam1 = STR, beam2 = STR: More flexibility is provided by setting the beams individually with the following allowed values: P (p), AP ($\bar{\text{p}}$), EM (e⁻), EP (e⁺), HE ($^4_2$He), LI ($^7_3$Li), C ($^{12}_6$C), O ($^{16}_8$O), CU ($^{64}_{29}$Cu), AU ($^{197}_{79}$Au), PB ($^{208}_{82}$Pb). Setting beam1 and beam2 will override the collider choice. It is the responsibility of the user to choose a compatible (nuclear) PDF set for each beam in the RUN block described below. The centre-of-mass energy is specified for the nucleon–nucleon system. In case of an asymmetric beams setup (beam1 $\neq$ beam2), NNLOJET will further perform the appropriate longitudinal boost of the hadronic centre-of-mass system. The collider coordinate system is then chosen such that beam1 is in the positive $z$ direction.

The collider energy is provided by setting either one of the following options:

- sqrts = DBLE [**required**]: definition of centre-of-mass energy;

- Ebeam1 = DBLE, Ebeam2 = DBLE: definition of energy of individual beams instead of specifying sqrts, allows for asymmetric collisions. Setting Ebeam1 and Ebeam2 will override the sqrts value.

The standard jet reconstruction algorithms are implemented directly in NNLOJET and can be specified with the following options:

- jet = <algo>[<options>]: specification of jet algorithm; the option in DBLE specifies the jet-resolution parameter. The algorithm can be chosen with the following syntax:

    - anti-$k_T$ algorithm [131]: antikt[DBLE];
    - Cambridge–Aachen algorithm [132] (longitudinally invariant): ca[DBLE];
    - $k_T$ algorithm [133,134] (longitudinally invariant): kt[DBLE];
    - Durham algorithm [135] (for e⁺e⁻ collisions): durham[DBLE];
    - Jade algorithm [136] (for e⁺e⁻ collisions): jade[DBLE];
    - No algorithm required: none;

- jet_exclusive = BOOL: jet exclusive flag, specifying whether events with more jets than required should be rejected (default: .false.);

- jet_recomb = STR: jet-recombination scheme: V4, to combine jets by adding the 4-vector of the momenta, and ET, to combine jets with $E_T$ as the weights (default: V4).

Several optional settings in the PROCESS block are only relevant for specific classes of processes:

- V_NC = STR: restriction on virtual neutral-current gauge-bosons. Options for STR:

    - ZGAMMA (default): both Z bosons and photons are allowed in the amplitudes;
    - GAMMA: only photons, useful for example in DIS processes;
    - Z: only Z bosons are allowed;

- L_NC = STR: decay mode of Z bosons decay to leptons. Values:

    - NU: decay to neutrinos (single family) $Z \to \nu \bar{\nu}$;
    - L (default): decay to charged leptons (single family) $Z \to \ell^+ \ell^-$.

- decay_type = INT: definition of heavy-particle decay types used in the phase-space generator. It is set automatically for each process and unstable particle, but can be overridden using the following INT values:

- 0: use $dM^2/M^2$;
- 1: use resonance parametrization;
- 2: narrow width inserted, delta function;
- 3: on-shell heavy particle, no decay.

The isolation of photons and leptons is defined using the following options:

- `photon_isolation`/`lepton_isolation`: define the isolation of non-QCD particles, as documented in detail in [113]. It allows the combination of several isolation prescriptions that must be simultaneously fulfilled, using the syntax
  `<isolation>` = `<algo1>`[`<params1>`] + `<algo2>`[`<params2>`] + ... .
  For lepton isolation, only fixed-cone isolation is supported. For photon isolation, the following isolation prescriptions are available:

  - `none`: no isolation required;
  - `fixed`: fixed-cone isolation. The non-QCD particle with transverse momentum $p_T$ is isolated if the hadronic transverse energy in a cone of size $R_{\text{cone}}$ does not exceed

    $$E_T^{\max} = E_T^{\text{const}} + \epsilon p_T \ . \tag{9}$$

    Parameters:
    * `R_cone`: isolation-cone size $R_{\text{cone}}$;
    * `ET_const`: $E_T^{\text{const}}$;
    * `ETmax_epsilon`: $\epsilon$;

  - `smooth`: smooth-cone isolation [110]. The non-QCD particle with transverse momentum $p_T$ is isolated according to a profile function $\mathcal{X}(R, n)$ that permits a decreasing amount of hadronic energy $E_T^{\text{smooth}} = \mathcal{X}(R, n) E_T^{\max}$ within sub-cones of radius $R < R_{\text{cone}}$ and $E_T^{\max}$ as defined in (9):

    $$\mathcal{X}(R, n) = \left[ \frac{1 - \cos(R)}{1 - \cos(R_{\text{cone}})} \right]^n \ .$$

  - `hybrid_smooth`: hybrid isolation. A variant of the smooth-cone isolation defined in [137]. The following parameters can be specified:
    * `ET_const`: $E_T^{\text{const}}$;
    * `ETmax_epsilon`: $\epsilon$;
    * `R_cone`: the smooth-cone size or the outer fixed-cone size in hybrid isolation;
    * `n`: parameter $n$ in profile function;
    * `R_inner`: inner smooth-cone size in case hybrid isolation is used, where `R_inner`<`R_cone` must be satisfied;

  - `democratic`: democratic isolation. The jet clustering procedure includes the non-QCD object, which is then called isolated if it carries more than a fraction $z_{\text{cut}}$ of the transverse energy of the jet it is assigned to. Parameters:
    * `zcut`: threshold of transverse-energy fraction between the photon and the jet. If the transverse energy fraction is larger than `zcut`, the clustered particles are considered as one photon; if the fraction is smaller than `zcut`, the clustered particles are considered as one jet.

- `photon_recomb` = `DBLE`: angular distance $R$ below which two photons are recombined;

- `photon_fragmentation` = `STR`: definition of photon fragmentation setup. The parameter specifies the parametrization of the photon fragmentation functions:

    - `ALEPH`: a simple parametrization obtained from a one-parameter fit by ALEPH [138];
    - `BFG0`, `BFGPERT`: BFG0 [114];
    - `BFG1`, `BFGI`: BFGI [114];
    - `BFG2`, `BFGII`: BFGII [114].

Finally, several `PROCESS` settings are relevant only for DIS:

- `pol` = `DBLE`: polarization fraction of incoming lepton beam (default: 0.0);

- `lab_ordereta` = `BOOL`: a flag to order jets in the laboratory frame in $\eta$ (default: `.false.`);

- `dis_eventshapes` = `DBLE`: minimum value of hadronic energy in an event for DIS event shapes (default: 0.0);

- `eprc` = `BOOL`: switch to use a running electromagnetic coupling evaluated at the scale $Q^2$ (default: `.false.`);

- `dis_frame` = `STR`: definition of the DIS frame, with the following options:

    - `FIXT`: fixed target system;
    - `BREIT`: Breit frame;
    - `LAB`: the laboratory frame.

    The default is `BREIT` for `epLJJ`, `epNJJ`, `epNbJJ` and `LAB` for `epLJ`, `epNJ`, `epNbJ`.

### 5.3.2 Run block

This block defines technical settings for the NNLOJET execution. Runs in NNLOJET consist of two phases:

- The `warmup` phase aims to construct VEGAS grids that are adapted to the integrand at hand (event-selection cuts, regions of larger weights etc.). The VEGAS grids are initialized as uniform distributions in the raw random variables, and the calculation proceeds in multiple iterations. After each iteration, the grids are adapted according to the accumulated cross section for importance sampling. The events sampled during the warmup phase *do not* enter the final result. For performance reasons, histogram filling and scales beyond the central scales are deactivated.

- The `production` phase performs the Monte Carlo integration for the total and differential cross sections according to the VEGAS grid that was adapted during the warmup. The VEGAS grids are frozen at this stage and will no longer be updated.

To run NNLOJET, at least one of the two phases needs to be activated. The `production` phase can only be performed if a dedicated VEGAS grid was produced by a `warmup` phase beforehand. When the `nnlojet-run` workflow is used to run NNLOJET, the `warmup` and `production` phases are automatically handled and can be omitted from the runcard.

The general syntax for the `RUN` block is:

```
RUN <run_name>
  <option1> = <value1>[<parameters1>]
  <option2> = <value2>[<parameters2>]
  ...
END_RUN
```

The field `<run_name>` specifies the user-defined name for the run and it is used to name the output files. The `<run_name>` for a `production` phase must be the same as the respective `warmup` phase for the VEGAS grids to be properly loaded. The options are:

- `PDF = <PDF_set>[<member>]` [**required**]: specification of a PDF set according to the naming convention of LHAPDF [67]. For asymmetric colliders, separate PDFs can be specified for each beam by instead providing `PDF1` and `PDF2` individually. The optional `<member> = INT` parameter specifies the PDF member. It defaults to `0`, i.e. the central member of the PDF set. Note that the PDF is required also for electron-positron collisions, for the running of $\alpha_s$;

- `iseed = INT`: specification of the seed for the random number generator (default: 1).;

- `warmup = <ncall>[<niter>]`: specification of warmup run settings. `<ncall> = INT` indicates the number of integrand evaluations per iteration and `<niter> = INT` the number of iterations. Instead of `<niter>`, it is also possible to specify `auto`, which will automatically distribute the total number of `<ncall>` evaluations of the integrand into a staggered batch of iterations (at least one of `warmup` or `production` needs to be specified);

- `production = <ncall>[<niter>]`: specification of production run settings; in particular, `<ncall> = INT` indicates the number of integrand evaluations per iteration and `<niter> = INT` the number of iterations (at least one of `warmup` or `production` needs to be specified);

- `tcut = DBLE`: technical cutoff defined relative to the squared partonic centre-of-mass energy $\hat{s}$. Events with any invariant smaller than `tcut`·$\hat{s}$ are rejected in the phase-space integration. The technical cut must be chosen considerably smaller than the smallest scale that is resolved by the final-state definition (default: `1d-8`);

- `reset_vegas_grid = BOOL`: switch for overwriting the local VEGAS grid; when set to `.true.`, the warmup run will be started with a new uniform VEGAS grid; when set to `.false.`, the warmup run will continue starting from the previous VEGAS grid if one exists (default: `.false.`).

NNLOJET also provides some optional optimization settings which can speed up the calculation when used properly. They are in general only intended for expert users. The following optimization settings are available:

- `angular_average = BOOL`: switch to average over azimuthal angle rotations in unresolved regions, required to make antenna subtraction terms strictly local (default: `.true.`);

- `cache_kinematics = BOOL`: switch to cache all kinematics of mapped momentum sets (default: `.false.`);

- `scale_coefficients = BOOL`: switch to internally use a decomposition of weights into scale coefficients of scale logarithms. Necessary for scale overrides (default: `.false.`);

- `multi_channel = <N>`: option for multi-channel adaption over individual channels. For the definition of channels, see 5.3.7. The parameter `<N>` is of type `INT`. If `<N>` is 0, every existing channel is evaluated in each sampled phase-space point. Multi-channeling is activated by selecting a non-zero `<N>`, leading to a selective evaluation of different channels by distributing events among them. An importance sampling is applied by evaluating those channels more often which have a larger contribution to the cross section. The

| | Up | Charm | Top |
|---|---|---|---|
| Quarks (Up type) | U | C | T |
| Quarks (Down type) | Down | Strange | Bottom |
| | D | S | B |
| Leptons | Electron | Muon | Tau |
| | EL | MU | TAU |
| Bosons | Higgs | W-boson | Z-boson |
| | H | W | Z |
| Others | Proton | | |
| | PROTON | | |

Table 1: Acronyms for particles to be used as `<particle>` of attributes.

strategy for dividing the channels into groups is determined by the value of `<N>`. If `<N>` is positive, channels will be grouped such that every phase-space point will evaluate a subset of `<N>` channels. If `<N>` is negative, the total number of groups is specified, i.e. the integrator will attempt to divide active channels into |`<N>`| equal-sized groups (default 0);

- `lips_reduce` = `BOOL`: switch for the reduced phase-space mode, a variant of the phase-space generator that exploits the permutation symmetries of a specific process (default: `.false.`).

### 5.3.3 Parameters block

The `PARAMETERS` block defines the Standard Model input parameters of the calculation. The general syntax follows the pattern:

```
PARAMETERS
  <variable>[<spec>] = <value>
  <option> = <value>
  ...
END_PARAMETERS
```

The attributes of particle `<particle>` are set with the syntax `<attribute>[<particle>]`. `<particle>` may assume any acronym from Table 1 as its value. The possible attributes are listed in the following, with the default value taken from [139]:

- `mass`[`<particle>`] = `DBLE`: the mass of the particle, which can be set for the massive particles in the Standard Model and the proton. The mass values correspond to the mass scheme defined below;

- `width`[`<particle>`] = `DBLE`: the decay width of the particle. Z, W, H and top quark are supported. The widths correspond to the scheme defined below.

Further parameters and prescriptions can be set as follows:

- `hard_photon_alpha0` = `BOOL`: switch to use $\alpha_0$ as the electroweak coupling for resolved photons in photon-production processes (default: `.false.`);

- `scheme`[`<scheme>`] = `STR`: the specification of schemes in mass, width and electroweak coupling. The options for `<scheme>` are:

- **MV**: define the mass scheme for electroweak gauge bosons. Possible values are:
  - **OS** (default): on-shell scheme;
  - **POLE**: pole scheme;
  - **MSBAR**: $\overline{\text{MS}}$-scheme;

- **GV**: define the width schemes for electroweak gauge bosons. Possible values are:
  - **NWA**: for the narrow-width approximation;
  - **CMS**: for complex-mass scheme;
  - **NAIVE** (default): for naive fixed-width scheme;
  - **RUN**: for running on-shell scheme;
  - **POLE**: for pole scheme;

- **ALPHA**: the electroweak coupling prescription. Possible values are:
  - **GMU** (default): $G_\mu$ scheme, derived from the Fermi constant and the Z and W masses (override default value with **GF** = **DBLE**);
  - **FIX**: fixed scheme (override default value with **alpha** = **DBLE**);
  - **ALPHA0**: $\alpha_0$ scheme (override default value with **alpha0** = **DBLE**);
  - **ALPHAMZ**: $\alpha(M_Z)$ scheme (override default value with **alphaMZ** = **DBLE**);

- **CKM** = **STR**: the specification of the CKM matrix. Users can change the entries by modifying `driver/core/CKM.f90` if necessary. The available options are:

  - **IDENTITY** (default): diagonal CKM matrix;
  - **FULL**: Wolfenstein parameters [140];
  - **CABIBBO**: Cabibbo mixing angle only [140];
  - **MCFM**: the default setup of MCFM.

### 5.3.4 Selectors block

In NNLOJET, two types of selection criteria are distinguished: object-level and event-level criteria. Object-level criteria may be used to impose conditions on the reconstructed objects, i.e. jets and photons. For example, they allow to select jets with a minimum transverse momentum or within a specific rapidity range. Any object that does not fulfill these conditions is omitted from the event record.

Event-level criteria define which event candidates should be used to calculate cross sections and fill histograms. If an event-level criterion is not fulfilled, the entire event candidate is discarded. Typical examples are a minimum number of jets per event, additional constraints on the leading jet, or constraints on leptons.

The **SELECTORS** block specifies the object-level and event-level selection criteria. It is encapsulated by

```
SELECTORS
  ! put selectors here
END_SELECTORS
```

It can be left empty if the process is well-defined (corresponding to a fully-inclusive event selection) or filled with any number of selector statements of the form:

```
<type> <observable> min=<val_min> max=<val_max>
```

- The `<type>` can be either `select` or `reject`. In case partially valid observables are expected, one can use `select_if_valid` or `reject_if_valid`. An event is selected only if it passes all `select` statements and is rejected by none of the `reject` statements.

- The `<observable>` is specified using a pre-defined string identifier. Possible values for a given process can be checked using the `--listobs` query of the NNLOJET command. Object-level cuts are listed in the `photon cuts` and `jet acceptance cuts` sections, while event-level selectors are listed in the `process-specific observables` section.

- It is only necessary to specify at least one of the two assignments for `min` and `max`. If unspecified, the other boundary defaults to $\pm\infty$. If both are specified, the order of the assignments is irrelevant.

Selector statements can be given in any order. Since each selector will internally generate a new object, it is better to have as few lines as possible. It is therefore preferable to write

```
select ylp min = -5 max = +5   ! --> accept [-5, +5]
```

or

```
select abs_ylp max = +5        ! --> accept |y| < 5 => [-5, +5]
```

instead of

```
select ylp min = -5            ! --> accept [-5, +infinity]
select ylp max = +5            ! --> accept [-infinity, +5]
```

It is possible to define `OR` groups, where all the selector statements inside the group have 'or' relations. The groups are encapsulated by

```
OR
  ! put selectors here
END_OR
```

It is not possible to use `OR` groups on object-level selectors.

A common example use case of the `reject` type in the selectors is the exclusion of the barrel–endcap gap region that can be implemented as follows:

```
select abs_ylp max = +5                        ! global selector
reject abs_ylp min = +1.4442 max = +1.560   ! barrel-endcap
```

which is more intuitive and cleaner than the alternative only using `select` and an `OR` group:

```
OR
  select ylp min = -5       max = -1.560
  select ylp min = -1.4442  max = +1.4442
  select ylp min = +1.560   max = +5
END_OR
```

### 5.3.5  Histograms block

The `HISTOGRAMS` block specifies all the histograms to be filled during the run. It is encapsulated between the lines

```
HISTOGRAMS
  ! put histograms here
END_HISTOGRAMS
```

It can be populated with histogram statements of the following two types

```
<observable> nbins=<val_nbins> min=<val_min> max=<val_max>
<observable> [edge0,edge1,edge2,...,edgeN]
```

- The first version corresponds to a histogram with uniformly-space (linear or logarithmic) bins, whereas the second version specifies the bin boundaries as a list and allows for flexible bin edges.

- The `<observable>` is specified using the unique string identifier and events giving rise to an invalid attribute (see Sect. 5.2 for the definition) are skipped in the accumulation.

- As a decorator to the `<observable>` specification, it is possible to specify a file-name override using the notation `<observable> > <dat_name>`. If this specification is omitted, the file name will use the observable name by default. This option is needed if there exist multiple histograms that bin the same observable (see example below) in order to avoid file-name clashes.

- For cross sections, the special name `cross` can be used instead of the histogram statements above. NNLOJET will *always* register such a histogram by default so that user-defined cross section statements must always be of the form `cross > <dat_name>` with a file-name override. It should be noted that histograms keep track of overflows such that the cross section can also be calculated from the histogram contents by summing the bins and the overflow. It is therefore rarely necessary to define a cross section histogram explicitly.

Every histogram statement can further be supplemented by optional settings. The available options are:

- `fac = <rwg_fct>`: multiply all weights passed to the histogram with a generic reweighting function following the syntax described in Sect. 5.2.1;

- `mu0 = (DBLE * )<observable>`: override scale. This options sets the both factorization and renormalization scale to the specified value. Only available if `scale_coefficients = .true.` is set in the `RUN` block;

- `binning = STR`: switch between linear and logarithmic binning, in case equal-sized bins are used. Possible values are:

  - `LIN` (default): linear binning;
  - `LOG`: logarithmic binning;

- `output_type = INT`: specify the luminosity-breakdown type: a value of 0 will only output the numbers for the total (default), while a value of 1 will in addition provide numbers broken down into separate partonic luminosities: (*e.g.* $\bar{q}\bar{q}$, $g\bar{q}$, $q\bar{q}$, $\bar{q}g$, $gg$, $qg$, $\bar{q}q$, $gq$, $qq$). See Sect. 6.3 for further description of the output data files;

- `cumulant = INT`: option for cumulant distributions with possible values:

  - 0 (default): standard differential distribution $d\sigma/d$`<obs>`;
  - -1: cumulant distribution $\int_{o_{\min}}^{o_i} d\sigma/d$`<obs>`;
  - +1: cumulant distribution $\int_{o_i}^{o_{\max}} d\sigma/d$`<obs>`.

  The values $o_{\min}$ and $o_{\max}$ respectively correspond to the lower and upper bounds of the histogram and the user must ensure that it covers the desired kinematic range. Observable values outside of this range are ignored in the cumulant output and are instead collected in an overflow bin.

A further optional specification is given by histogram selectors which can be assigned right below a histogram definition as follows:

```
<observable> ...
HISTOGRAM_SELECTORS
  ! put selectors here
END_HISTOGRAM_SELECTORS
```

with the selector syntax defined in Sect. 5.3.4 but without the support of `OR` blocks. This allows to generate multi-differential distributions in a flexible way as follows:

```
ptz > ptz_bin1 nbins=75 min=0 max=150
HISTOGRAM_SELECTORS
  select abs_yz max=0.3
END_HISTOGRAM_SELECTORS

ptz > ptz_bin2 nbins=30 min=0 max=90
HISTOGRAM_SELECTORS
  select abs_yz min=0.3 max=1
END_HISTOGRAM_SELECTORS

ptz > ptz_bin3 [0,10,15,20,25,30,40,50,60,80,100]
HISTOGRAM_SELECTORS
  select abs_yz min=1 max=2
END_HISTOGRAM_SELECTORS
```

Here, the modified file names (`> ptz_bin<i>`) are needed in order to avoid file name clashes. Furthermore, we can choose different `ptz`-binnings for each `yz` bin. The histogram file will correspond *precisely* to what is specified in the runcard, for the example above this means

$$\texttt{ptz\_bin1}: \qquad \left.\frac{\mathrm{d}\sigma}{\mathrm{d}p_{\mathrm{T,Z}}}\right|_{|y_Z|<0.3} \neq \frac{\mathrm{d}^2\sigma}{\mathrm{d}p_{\mathrm{T,Z}}\,\mathrm{d}y_Z} \quad \text{(in the first } y_Z \text{ bin).} \tag{10}$$

In order to obtain the double-differential distribution on the right-hand-side of this expression one has to further divide the histogram contents by the bin-width for $y_Z \in [-0.3, 0.3]$ which is 0.6 in this case.

The way invalid observables (see Sect. 5.2 for the definition) are treated within the histogram selectors deserves special care. Once an observable is used in a `select` or `reject` statement within the `HISTOGRAM_SELECTORS` block, it is *implicitly assumed* that it must be valid for all events that are considered for the histogram filling. This means that if an invalid observable is encountered for such a specification, the weight associated with that event is discarded in the accumulation of the histogram. If this is not the desired behaviour, it is instead possible to use the `select_if_valid` or `reject_if_valid` specifications that allow to ignore the selector in case the observable is evaluated with an invalid attribute.

Some observables are defined in a composite manner, i.e. they can potentially induce *multiple* bookings to the same histogram from different observables. For this purpose, NNLOJET defines the concept of a composite histogram that follows the following syntax:

```
COMPOSITE > <dat_name> ... <opt_comp>
  <obs1>  <opt1>
  <obs2>  <opt2>
  ...
END_COMPOSITE
```

where `...` at the top can be the fixed-bin or the variable-bin specification and `<opt_comp>` the optional specifications both described above. The composite histogram will be filled with the sum of all the sub-histograms defined inside the block, where each can specify individual options, i.e. `fac` and `mu0` as described above. Both the composite histogram as well as the sub-histograms can further be supplemented by histogram selectors. In case the observable of the sub-histogram is invalid, it will simply be skipped in the accumulation.

A non-trivial example of this feature is the inclusive-jet distribution in a specific rapidity bin and a jet-based scale setting such as the jet-$p_T$. In this case, we must register as many sub-histograms as the maximum possible number of jets (invalid observables will be skipped),

each dressed with a selector block and further ensure that for each sub-histogram, the weight is re-assembled for the value of the respective jet $p_T$ as the scale:

```
COMPOSITE > ptj0_ybin0 [0,10,20,50,100,200,300,500,700,1000]
  ptj1   mu0=ptj1
  HISTOGRAM_SELECTORS
    select abs_yj1 min=0 max=.5
  END_HISTOGRAM_SELECTORS
  ptj2   mu0=ptj2
  HISTOGRAM_SELECTORS
    select abs_yj2 min=0 max=.5
  END_HISTOGRAM_SELECTORS
  ptj3   mu0=ptj3
  HISTOGRAM_SELECTORS
    select abs_yj3 min=0 max=.5
  END_HISTOGRAM_SELECTORS
  ptj4   mu0=ptj4
  HISTOGRAM_SELECTORS
    select abs_yj4 min=0 max=.5
  END_HISTOGRAM_SELECTORS
END_COMPOSITE
```

### 5.3.6  Scales block

The `SCALES` block specifies all scale combinations that should be evaluated *simultaneously* during the run. It is encapsulated by the block:

```
SCALES
  ! put scales here
END_SCALES
```

and requires at least one scale specification to be present. The first entry will be considered the central scale that is also used for the VEGAS grid adaption. Up to six additional scale specifications can be supplied. A scale specification constitutes a line in the block that specifies the factorization scale $\mu_F$ (`muF`) and the renormalization scale $\mu_R$ (`muR`)

```
muF = <sclF>   muR = <sclR>
```

In case a process that contains a fragmentation function is run, the associated factorization scale ($\mu_A$) can be set separately using a pair `muF = [<sclF>,<sclA>]`, which otherwise will default to $\mu_A = \mu_F$ in case only a single scale is specified. The scale `<scl>` can be set in two ways:

- fixed scale: `<scl>` = `DBLE` in units of [GeV];

- dynamical scale: `<scl>` = `<obs>` or `DBLE*<obs>`, where `<obs>` is the unique `STR` identifier of an observable.

### 5.3.7  Channels block

In NNLOJET a *channel* constitutes a specific scattering process, uniquely determined by the external states and the crossing, further decomposed into gauge-invariant colour factors. For example, the leading-colour $\mathcal{O}(N_c^2)$ contribution to the scattering sub-process $gg \to q\bar{q}$ constitutes one *channel* of the LO dijet production process at hadron colliders.

When the workflow described in Sect. 6.1 is used to perform the calculation, the channel information will be automatically populated, such that the user does not need to specify them

manually and instead can keep this block empty. Therefore, the information provided here is intended for expert users and/or to assist in debugging the workflow calculations.

The `CHANNELS` block specifies all the channels to be computed during the run by giving a list of identifiers. The VEGAS integrand is defined as the sum of the channels specified in this block (optionally grouped according to the value of the `multi_channel` flag, see Sect. 5.3.2). The process-id number associated with a given channel can be found in the respective `driver/process/XYZ/selectchannelXYZ.f` source file of the associated process XYZ. In addition, the following abbreviations are defined to simplify the specification:

- `ALL`;

- `LO, NLO, NNLO`;

- `V, R`;

- `VV, RV, RR, RRa, RRb`.

The suffices `a` and `b` for the double-real (`RR`) contribution only exist for specific processes and correspond to a separation of the phase space into two disjoint parts with optimized sampling strategies. The general syntax for the channel block is as follows:

```
CHANNELS region = STR
  ! put list of process id's / wildcards here
END_CHANNELS
```

In addition to specifying the sub-process on individual lines, any number of (white-space separated) process id's can be given in a single line as well. The `region` option only matters for the double-real contribution and processes where a phase-space separation was introduced; allowed values in this case are: `a` for the `RRa` piece, `b` for the `RRb` piece, and `all` (default) to include both.

### 5.3.8 Single-line Optional Settings

In a runcard, the user is further able to specify single-line options independent of any blocks mentioned above. All single-line settings are optional.

**Reweight**

The re-weighting feature has a one-line syntax

```
REWEIGHT <rwg_fct>
```

where the specification of the reweighting function `<rwg_fct>` follows the syntax described in Sect. 5.2.1. This function is evaluated for every event and used to re-scale the weights returned to the VEGAS integrator. It can therefore be used to increase or decrease the relevance of certain kinematic regions in the VEGAS grid adaptation, allowing for a more uniform phase-space coverage for distributions that span over several orders of magnitude. The re-weighting does not apply to the event booking into histograms. Similarly, once the VEGAS grid is frozen, i.e. during the production stage of the calculation, this option has no impact on the produced output.

**PDF Interpolation**

The PDF interpolation option is a single-line setting to switch on a one-dimensional cubic spline interpolation (fixed $x$, interpolation in $\mu_F$), which aims to reduce the number of calls to LHAPDF. By default it is not activated but it can be switched on with the following syntax:

```
PDF_INTERPOLATION stepfac = DBLE
```

where the optional argument `stepfac` specifies the uniform spacing of the interpolation grid in the $\log(\mu_F)$ space as a multiplicative factor. The default value is 2.

# 6  Using NNLOJET

Following the installation steps outlined in Sect. 3.2, both the core executable (`NNLOJET`) as well as the Python workflow (`nnlojet-run`) are installed in the `bin` directory of the installation path. The main user-facing interface to NNLOJET is the workflow script `nnlojet-run`, which provides an automated way to set up and run calculations and is described in Sect. 6.1. The direct execution of the core executable is possible but requires a more in-depth knowledge of the inner workings of the program, and is therefore only recommended for expert users. Section 6.2 is therefore limited to a description of the output files that is produced by the core `NNLOJET` executable but will refrain from a detailed guide of a workflow based on it. The understanding of the output files is important in order to inspect individual runs submitted by the Python workflow in case issues are encountered. The main results are provided in the form of histogram files that are described in Sect. 6.3.

## 6.1  Job submission workflow

Every NNLOJET calculation (or *run*) is associated with a directory that contains all the necessary input and configuration files as well as the intermediate output and the final result. The `nnlojet-run` workflow is designed to manage the creation of the run directory, the execution of the NNLOJET core executable, and the retrieval of the final results in a user-friendly way. To this end, the workflow provides four separate sub-commands: `init`, `config`, `submit`, and `finalize`, which are described in the following. A short description of the commands and available options can be displayed by adding the `--help` command line argument:

```
$ nnlojet-run --help
$ nnlojet-run <sub-command> --help
```

### 6.1.1  `init`

The `init` sub-command initializes a new run directory with the necessary input and configuration files. It takes an NNLOJET runcard as input:

```
$ nnlojet-run [--exe <exe-path>] init [-o <run-path>] path/to/
    runcard
```

The command will create a new run directory at the current path named after the `<run-name>` specified in the input runcard. Optionally, the option `-o <run-path>` can be used to specify a location where the run directory should be initialized. The input runcard will be used to generate a template runcard for the calculation, which is saved as the file `template.run` in the run directory. The NNLOJET executable is automatically searched for in the system path but can also be set manually with the optional `--exe <exe-path>` argument.

The `init` command will first display the relevant references for the selected process that should be cited in a scientific publication using results obtained from the NNLOJET calculation; these references will also be saved as `.tex` and `.bib` files in the run directory. Subsequently, the user will be prompted to set several configuration options that are necessary for the execution:

- `policy` = `local`|`htcondor`|`slurm`: specifies the execution policy for how the NNLO-JET calculations should be submitted. Currently supported targets are local execution

on the current machine (`local`) or submission on a cluster using the job scheduling systems `slurm` or `htcondor`. The cluster submission requires additional settings that are documented below. (default: `local`)

- `order` = `lo`|`nlo`|`nlo_only`|`nnlo`|`nnlo_only`: specifies the perturbative order of the calculation to be performed. It is also possible to restrict the calculation to only a specific coefficient, `nnlo`=`lo`+`nlo_only`+`nnlo_only`. (default: `nnlo`)

- `target-rel-acc` = `DBLE`: specifies the desired relative accuracy on the total cross section evaluated with the chosen fiducial cuts and for the specified `order`. While the specification of a `REWEIGHT` function can be introduced to bias the phase-space sampling, the target accuracy is always defined with respect to the unweighted cross section. (default: `0.01`)

- `job-max-runtime` = `STR`|`DBLE`: specifies the maximum runtime for a single NNLOJET execution or a job on the cluster. The format is either a time interval specified with units (s, m, h, d, w), e.g. `1h 30m`, or a floating point number in seconds. (default: `1h`)

- `job-fill-max-runtime` = `yes`|`no`: specifies if the workflow should attempt to exhaust the maximum runtime that was set for each NNLOJET execution. This option is intended to fully utilize the wallclock time limit of cluster queues. (default: `no`)

- `jobs-max-total` = `INT`: specify the maximum number of NNLOJET executions that can be run in total. The workflow will terminate either once the target accuracy is reached or this limit on the total number of jobs is exhausted. (default: `100`)

- `jobs-max-concurrent` = `INT`: specifies the maximum number of NNLOJET executions that can be run simultaneously. If cluster submission is selected, this option specifies the maximum number of jobs that are simultaneously submitted to the cluster. (default: `10`)

If a cluster submission is selected, the user will be prompted to provide additional settings required for an execution on the respective cluster type:

- `poll-time` = `STR`|`DBLE`: specifies the interval in which the scheduler is queried for an update on the status of the submitted jobs. The format is either a time interval specified with units (s, m, h, d, w), e.g. `1h 30m`, or a floating point number in seconds. (default: 10% of the `job-max-runtime`)

- `submission-template` = `INT` (index from a list displayed to the user): the job submission on a cluster relies on a submission script for the job scheduler. The workflow provides a set of built-in submission templates that can be selected by the user. The chosen template will be copied as a file `<cluster>.template` to the run directory and can be modified to any specific needs. Note that it is up to the user to ensure that the time limits set by the submission script, e.g. as imposed by a wallclock time limit of special queues of the cluster, are consistent with the `job-max-runtime` setting.

All configurations are stored in the `config.json` file inside the run directory, however, it is strongly advised against modifying this file manually. Instead, all interactions with the run directory should proceed through the `nnlojet-run` interface. The configuration of the run can be changed at any time using the `config` sub-command described below.

### 6.1.2  `submit`

With a properly initialized run directory at `<run-path>`, the main calculation can be triggered using the `submit` sub-command:

```
$ nnlojet-run submit <run-path>
```

This calculation will employ the options that were set during the `init` step (or alternatively overwritten using the `config` sub-command). Some of the options can also be overridden temporarily for a single submission by passing the desired settings as parameters, e.g.

```
$ nnlojet-run submit <run-path> --job-max-runtime 1h30m \
  --jobs-max-total 10 --target-rel-acc 1e-3
```

A complete list of option overrides with a short description is provided via

```
$ nnlojet-run submit --help
```

Once the `submit` sub-command is executed, the workflow is started and the process remains active for the entirety of the calculation, spawning sub-processes to execute the core `NNLOJET` program. The full calculation is divided into separate components represented by the individual lines of Eqs. (6) and (7) and further decomposed into independent partonic luminosities. The workflow proceeds by first adapting the phase-space grids in a warmup phase, followed by the actual production phase as described in Sect. 5.3.2. The warmup phase proceeds in steps, iteratively increasing the invested statistics to gradually refine the adaption until either a satisfactory grid is obtained or the specified maximum runtime is reached. The production phase optimizes the allocation of computing resources into the different parts of the calculation to minimize the final uncertainty. To this end, it is necessary to obtain an estimate on the error and the runtime per event for each piece, which is initially determined from a single low-statistics *pre-production* run. As both the warmup and the pre-production stages are necessary to proceed to the main calculation, the associated jobs do not count towards the maximum number of jobs that was set for the submission and this part of the computation will not attempt to exhaust the available resources. During the main production stage, on the other hand, the workflow will continuously dispatch new jobs in an attempt to exhaust the specified concurrent resource limits and accumulate results until either the target accuracy is reached or the maximum number of jobs is exhausted. Upon termination of all jobs, a summary of the calculation is given by printing the final cross section number with its associated uncertainty. In case the target accuracy was not reached with the allocated resources, an estimate of the number of additional jobs required to reach the requested accuracy is provided. The final results of the calculation are saved to `<run-path>/results/final`. During the workflow execution, a summary table together with logging information will be displayed to the user, see Fig. 1, which is updated in real-time as the jobs are submitted and completed.

As the `submit` command continuously monitors running jobs and dispatches new ones, it is preferrable to keep it running for the entire duration of the calculation. If necessary, it can be stopped by pressing `Ctrl+C` and re-launched at a later time. In this case, the calculation will resume from the last saved state. Jobs that are already submitted to the cluster will continue to run. If the previous execution terminated exceptionally, e.g. due to a lost `ssh` connection, the workflow will automatically attempt to recover the state and prompt the user how to proceed with the resurrection of previously active jobs.

Inbetween executions of `submit`, the settings of a run can be modified, see the `config` sub-command below. This is especially useful to increase the target accuracy, or to increase the maximal number of jobs, in case too few where initially specified. It is also possible to change the perturbative order in this way, e.g. to compute the NNLO corrections to a run that was previously calculated to NLO.

### 6.1.3 `finalize`

All completed jobs inside a run directory `<run-path>` can be collected and merged into a final result using the `finalize` sub-command:

| # | LO | R | V | RR | RV | VV |
|---|----|----|----|----|----|----|
| 1 | PRD A[0] D[1] | PRD A[0] D[1] | PRD A[0] D[1] | WRM A[0] D[4] | **PRD** A[1] D[0] | PRD A[0] D[1] |
| 2 | PRD A[0] D[1] | **PRD** A[1] D[0] | PRD A[0] D[1] | PRD A[0] D[1] | WRM A[1] D[3] | PRD A[0] D[1] |
| 3 | - | PRD A[0] D[1] | PRD A[0] D[1] | PRD A[0] D[0] | WRM A[1] D[3] | PRD A[0] D[1] |
| 4 | - | PRD A[0] D[1] | PRD A[0] D[1] | PRD A[0] D[1] | PRD A[0] D[1] | PRD A[0] D[1] |
| 5 | - | PRD A[0] D[1] | PRD A[0] D[1] | PRD A[0] D[0] | PRD A[0] D[1] | PRD A[0] D[1] |
| 6 | - | PRD A[0] D[0] | PRD A[0] D[1] | PRD A[0] D[1] | PRD A[0] D[1] | PRD A[0] D[1] |
| 7 | - | - | - | PRD A[0] D[0] | PRD A[0] D[1] | PRD A[0] D[1] |
| 8 | - | - | - | **PRD** A[1] D[0] | PRD A[0] D[1] | PRD A[0] D[1] |
| 9 | - | - | - | PRD A[0] D[0] | **PRD** A[1] D[0] | PRD A[0] D[1] |
| 10 | - | - | - | **PRD** A[1] D[0] | PRD A[0] D[1] | PRD A[0] D[1] |
| 11 | - | - | - | PRD A[0] D[0] | PRD A[0] D[1] | PRD A[0] D[1] |
| 12 | - | - | - | PRD A[0] D[1] | - | - |
| 13 | - | - | - | PRD A[0] D[0] | PRD A[0] D[1] | PRD A[0] D[1] |
| 14 | - | - | - | PRD A[0] D[0] | - | - |
| 15 | - | - | - | PRD A[0] D[1] | WRM A[1] D[3] | PRD A[0] D[1] |
| 16 | - | - | - | **PRD** A[1] D[0] | WRM A[1] D[3] | PRD A[0] D[1] |
| 17 | - | - | - | PRD A[0] D[0] | WRM A[1] D[3] | PRD A[0] D[1] |
| 18 | - | - | - | **PRD** A[1] D[0] | - | - |
| 19 | - | - | - | PRD A[0] D[1] | - | - |
| 20 | - | - | - | PRD A[0] D[1] | - | - |
| 21 | - | - | - | WRM A[1] D[3] | - | PRD A[0] D[1] |
| 22 | - | - | - | PRD A[0] D[1] | - | PRD A[0] D[1] |
| 23 | - | - | - | **PRD** A[1] D[0] | - | PRD A[0] D[1] |
| 24 | - | - | - | WRM A[1] D[3] | - | PRD A[0] D[1] |
| 25 | - | - | - | WRM A[1] D[3] | - | - |
| 26 | - | - | - | **PRD** A[1] D[0] | - | - |
| 27 | - | - | - | PRD A[0] D[1] | - | - |

Figure 1: Example of the graphical interface to monitor the status of the calculation. Each column (LO, R, V, RR, RV, VV) represents an active component of the calculation. The indexed rows represent independent luminosity channels. In each cell of the table, the phase of calculation is indicated: warmup ("WRM") or production ("PRD"). The number $n_A$ of active jobs is indicated as A[$n_A$], which includes both pending and actively running jobs, while the number $n_D$ of completed jobs is indicated as D[$n_D$] . In case of unsuccessful NNLOJET executions, the cell will additionally include an entry F[$n_F$], indicating the number of failed jobs $n_F$.

```
$ nnlojet-run finalize <run-path>
```

It should be noted that the same combination procedure is also triggered at the end of the submit sub-command and it is therefore not necessary to explicitly execute a finalize after a submit.

The finalize sub-command facilitates a manual trigger to perform a re-combination of results with changed internal parameters of the merging algorithm. The main purpose of the merging algorithm is to combine the partial results of different parts of the calculation into a final prediction and to mitigate the impact from outliers. Potential issues pertaining to the treatment of outliers, as well as the approach employed in our implementation are presented in Ref. [141]. The underlying algorithm is controlled by a set of parameters that have sensible defaults but can also be overridden by the user. This is useful to check if the combine step is potentially introducing a systematic bias as well as to optimize the combination to obtain the best possible result from the raw data. The merging proceeds in steps with a dynamical termination condition, which are applied for each histogram separately on a *bin-by-bin* basis:

1. trim: apply an outlier-rejection procedure based on the inter-quartile-range to discard data points which are identified as severe outliers;

2. *k*-scan: on the trimmed dataset, merge pairs of (pseudo-)jobs in an unweighted manner into a new pseudo-job, which is statistically equivalent to running a single job with the

    combined statistics;

3. weighted combination: all pseudo-jobs are merged using a weighted average to further suppress outliers and the result is stored;

4. repeat steps 2 & 3 until `nsteps` results from weighted combinations are accumulated;

5. termination: check for a plateau in the last `nsteps` results; if no plateau is found, go back to step 2.

The core strategy is to combine as many individual jobs into pseudo-jobs as necessary to ensure that the resulting statistical uncertainty for the pseudo-job is reliable and therefore the weighted average can be safely taken in the next step to further suppress the impact of outliers.
    The parameters controlling the trimming step are:

- `trim-threshold` = `DBLE`: a threshold value for the distance from the median in units of the inter-quartile-distance beyond which a data point will be considered an outlier and is discarded. Larger values for this parameter reduce the number of data points that are discarded. (default = `3.5`)

- `trim-max-fraction` = `DBLE`: a safeguard on the maximal fraction of data points that are allowed to be trimmed away. For each bin in violation of this condition, its threshold value is dynamically increased until the ratio of trimmed data in that bin falls below this value. (default = `0.01`)

The termination condition of identifying a plateau (the $k$-scan) is controlled by the following parameters:

- `k-scan-nsteps` = `INT`: the number of pair-combination steps across which the pleateau is checked. (default: `2`)

- `k-scan-maxdev-steps` = `DBLE`: the tolerance for the plateau condition in units of the standard deviation. Across the last `k-scan-nsteps` steps of the $k$-scan, require that all points are mutually compatible within this tolerance. Note that the default value is smaller than 1 as the individual points from the $k$-scan are determined from the *same* dataset and thus are strongly correlated. (default = `0.5`)

The default values for this parameter can be set using the `config` sub-command described below. Alternatively, these options can also be overridden temporarily for a single `finalize` step by passing the desired settings as options, e.g.

```
$ nnlojet-run finalize <run-path> --trim-threshold 5 \
  --k-scan-nsteps 3 --k-scan-maxdev-steps 0.1
```

which is documented in the corresponding help output: `nnlojet-run finalize --help`.

### 6.1.4  `config`

All default configuration settings can be changed at a later time using the `config` sub-command:

```
$ nnlojet-run config <run-path>
```

This will open an interactive prompt that allows the user to change the configuration settings that were set during the `init` step. The merge settings used for the `finalize` sub-command can instead be overwritten using:

```
$ nnlojet-run config <run-path> --merge
```

In addition, advanced settings of the workflow can be viewed and set with

```
$ nnlojet-run config <run-path> --advanced
```

which provides, e.g. the possibility to change the seed offset of the workflow.

### 6.1.5 Run directory

The run directory is structured as follows:

- `NNLOJET_references_<PROC>.[bib|tex]` for the references to be cited in publications;

- `config.json`: contains all the configuration settings for the run;

- `db.sqlite`: a database file that contains the status of the run (process and job information);

- `log.sqlite`: a database file that contains the logging information of the submission;

- `template.run`: the template runcard for the calculation;

- `raw` directory: contains the raw output of the NNLOJET runs saved in two subdirectories (`warmup`, `production`) with the individual job results;

- `results` directory: contains the final results of the calculation;

  - `final` directory: contains the final results in the form of data files `<order>.<obs>.dat` for all complete orders;

  - `merge` directory: contains the result of the currently active order;

  - `parts` directory: all intermediate combined results separately for each part of the calculation;

The file format of the histogram data files is described below in Sect. 6.3.

## 6.2 Direct execution

The direct execution of the core `NNLOJET` executable produces output files with the following naming conventions:

- Log files: `<proc>.<run>.s<seed>.log`;

- Grid files (machine readable):
  `<proc>.<run>.y<t-cut>.<cont>`;

- Grid files (human readable):
  `<proc>.<run>.y<t-cut>.<cont>_iterations.txt`;

- Histogram files:
  `<proc>.<run>.<cont>.<obs>.s<seed>.dat`;

where `<proc>` denotes the process name, `<run>` the run name, `<seed>` the seed number, `<cont>` the contribution (LO, R, V, ...), and `<obs>` the observable name.

## 6.3 Histogram data files

The data files produced by NNLOJET are in a simple text-based format with units of [fb] for cross sections and [GeV] for dimensionful quantities. The first $n_x$ columns of the files are reserved for the binning information, if necessary:

- cross section data files have no bins and thus $n_x = 0$;

- differential distributions have $n_x = 3$ for the lower edge, centre, and upper edge of each bin;

- cumulant distributions have $n_x = 1$ for the integration limit for the cumulant.

The remaining columns are composed of a series of pairs of numbers that represent the value and the associated Monte Carlo integration uncertainty. First, these pairs are written out for all the active scales, as specified in the `SCALES` block and its order. Depending on the `output_type` specified for the histograms, this is optionally repeated with the same pattern for the separate partonic luminosities.

A description of the individual columns is also provided as a comment line starting with `#labels`. In addition, the cross section contribution from events that fall outside of chosen histogram range are collected and written out in the `#overflow` line following the same column structure.

# 7 Conclusion

This paper accompanies the open-source release of the parton-level event generator NNLOJET. The NNLOJET code is distributed under the GPLv3.0 license and can be downloaded from HEPForge: http://nnlojet.hepforge.org.

NNLOJET is steered through a runcard whose syntax is documented in this paper. Some example runcards are distributed with the release and more examples with sample output data are provided via http://nnlojet.hepforge.org/examples.html. The NNLOJET program can either be executed directly or through a workflow script that is part of the distribution and which dynamically adapts the usage of computing resources towards a pre-specified target accuracy. We described the usage of the NNLOJET executable and the `nnlojet-run` workflow script in detail.

In its current release (v1.0.0), NNLOJET contains a range of lepton- and hadron-collider jet production processes of $2 \rightarrow 2$ and $1 \rightarrow 3$ kinematics. More processes and features will be added in future releases and the code documentation on http://nnlojet.hepforge.org will be kept up to date to reflect those changes. User feedback and bug reports should be addressed to: nnlojet-support@cern.ch.

# Acknowledgements

The NNLOJET project has been enabled by substantial support throughout the past decade from the European Research Council (ERC) through grant agreements 340983 (ERC Advanced Grant MC@NNLO) and 101019620 (ERC Advanced Grant TOPUP), from the Swiss National Science Foundation (SNF) under contracts 200020-162487, 200021-172478, 200020-175595, 200021-197130, 200020-204200 and 200021-231259, from the Swiss National Supercomputing Centre (CSCS) under project IDs ETH5f and UZH10, from the Royal Society, from the UK Science and Technology Facilities Council (STFC) under contracts ST/G000905/1, ST/P001246/1, ST/T001011/1 and ST/X000745/1, and from the National Science Foundation of China (NSFC) under contract No.12475085, No.12321005 and No.1223500. MM is supported by a Royal Society Newton International Fellowship (NIF/R1/232539). JP acknowledges Fundação para a Ciência e a Tecnologia (FCT) for its financial support through the programatic funding of R&D units (contract UIDP/50007/2020) and under project CERN/FIS-PAR/0032/2021.

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
