# Peer review of "NNLOJET: a parton-level event generator for jet cross sections at NNLO QCD accuracy"

_SciPost Physics Codebases_

## Round 1 · Referee Report · Anonymous (Referee 1) · 2025-7-25

Report

This paper presents the first public release of NNLOJET, a parton-level Monte Carlo framework for computing jet cross sections at NNLO QCD accuracy using the antenna subtraction method. It supports a wide range of collider processes (hadron-hadron, lepton-hadron, and electron-positron) and provides a highly modular and flexible setup for defining fiducial observables and running differential calculations.

The release of this code is timely and important. It fills a notable gap in the community by making previously private NNLO implementations available in open-source form. The paper provides detailed documentation of the framework’s structure, runcard syntax, and supported processes.

The validation strategy outlined, pointwise subtraction tests, pole cancellation, and technical cut independence, is methodologically sound and aligns with best practices for NNLO QCD computations. However, for transparency and reproducibility, it would be beneficial if future versions of the documentation included explicit numerical validation outputs, such as:

(a) Subtraction term vs. matrix-element ratios near infrared limits
(b) Numerical pole cancellation checks
(c) Sample cross sections with uncertainty estimates

Additionally, while the paper refers to an example page, it would improve usability to more clearly highlight this resource early in the text or include a minimal working run card example directly in the manuscript.

Despite these minor suggestions, the scientific content is strong, and the software represents a valuable and technically robust contribution to high-energy theory.

I recommend acceptance with no required changes. The suggestions above are offered for future improvement, but do not warrant delaying publication.

Recommendation

Publish (easily meets expectations and criteria for this Journal; among top 50%)

---

## Round 1 · Referee Report · Anonymous (Referee 2) · 2025-9-4

Disclosure of Generative AI use

The referee discloses that the following generative AI tools have been used in the preparation of this report:

Claude (Opus 4.1). After finishing the report I asked for a report from Claude to see that I had not missed anything critical. The report mostly agreed with mine and I have kept the wording of my own report exactly.

Strengths

  1. First open source release of the antennae subtraction framework NNLOJET. Although not all processes presented by the group are included, the list is very comprehensive, in particular given that no other public codes can handle hadron-hadron processes with jets. As such this is a much needed piece of software in the field.
  2. The paper is very clear, colour-coded and discusses the program in great detail. In that sense the documentation is complete.
  3. The code compiles following the instructions and is relatively easy to use following the manual.

Weaknesses

  1. Although benchmarking tests are discussed they are not provided in the code or manual.
  2. There is no discussion of the code/method in comparison to other methods. In particular it would be useful for a user to have a sense of timings for the various processes to reach a certain level of precision. This can help a user make informed decisions about when to use NNLOJET and when to use other public tools like MATRIX or older/dedicated tools.
  3. Delivering the code as a tarball on hepforge is a bit old-fashioned and severely limits community interaction. Currently no changelog available which is not good code practice.

Report

Let me first say that the publication of the NNLOJET code is highly timely and fills a huge gap in the community. As such the authors deserve commendation for undertaking the publication, which is no small feat. Overall the documentation is very clear and well-structured, the coding is of excellent quality.

However, I do think that the code and paper are lacking in certain respects. Given that the NNLOJET code is expected to find wide-spread use both within the experimental and theoretical communities, I think it is important that they are addressed, to maintain the highest coding practices. I therefore recommend accepting the paper after some minor(ish) revisions.

Requested changes

First two general comments:

  1. The authors should provide some benchmarks to comply with the acceptance criteria of the journal. I suggest two additions: 1. a table with cross sections for all the processes that are implemented (with some sensible jet cuts and photon isolation when required), together with their uncertainties and required CPU wall time. Secondly a representative example for the points-test/pole cancellation is also needed to give the user a sense of the residual uncertainties associated with the technical cut. This should address the two first weaknesses above.
  2. Although it is up to the authors to decide how they host their code, I strongly suggest moving to a platform that allows for easier interaction and version control, eg. github/gitlab. As a minimum however, the hepforge page and code bundle need a changelog so that users can keep track of updates to the code (I see that there is a v1.0.2 of the code but have no way to know what changed, and also what happened to v1.0.1?).

Then a few specific comments in the text:

p.4 The list of public codes capable of NNLO is not complete. To mind comes fewz, dynnlo/dyturbo, hnnlo, provbf, and sherpa (which since v3 has public NNLO). Some of them are older/more specialised but the NNLOCAL code that is already listed can as far I know only do Higgs production, so is easily as specialised. p.7 NNLOJET comes bundled with HPLOG, TDHPL, and CHAPLIN. All three programs are published under the CPC license (https://www.elsevier.com/about/policies-and-standards/open-access-licenses/elsevier-user/cpc) or no license at all which is not compatible with GPLv3. Given that Gehrmann is an author of the two first I think it can be assumed that permission was obtained for those, but have the authors obtained permission to publish CHAPLIN under GPLv3? p.7 When discussing compilers it would be useful to know 1) what versions of Fortran/C++ compilers are known to work (GNU compilers, Intel compilers?) and 2) what kind of standard is needed FOrtran2003/2008... C++14 or later etc? p.8 When the authors make a reference to the examples I think it would be useful to provide a simple example in detail here -- also to comply with the second acceptance criterium. p.9 When discussing the points tests, is it clear how the distribution approaches zero when the control parameter is decreased (linearly, quadratically?). Is the control parameter dimensionfullor not? At this point it would also be useful to have a discussion of the advantages/limitaions of antennae subtraction compared to other subtractions and slicing. p. 30 The authors discuss the intricate merging procedure that they apply. In step 3. they take a weighted average. This is known to be a biased estimator unless errors are strictly gaussian. How exactly do you assess the bias? Also, have you considered winzorising in step 1. instead of rejection?

Recommendation

Ask for minor revision

---

## Editorial Decision

awaiting_resubmission